# Targeting the Stromal Pro-Tumoral Hyaluronan-CD44 Pathway in Pancreatic Cancer

**DOI:** 10.3390/ijms22083953

**Published:** 2021-04-12

**Authors:** Tomas Koltai, Stephan Joel Reshkin, Tiago M. A. Carvalho, Rosa A. Cardone

**Affiliations:** 1Via Pier Capponi 6, 50132 Florence, Italy; 2Department of Bioscience, Biotechnology and Biopharmaceutics, University of Bari, 70126 Bari, Italy; tiagomac94@gmail.com (T.M.A.C.); rosaangela.cardone@uniba.it (R.A.C.)

**Keywords:** pancreatic ductal adenocarcinoma, pancreatic cancer stroma, bromelain, 4-methylumbelliferone, pirfenidone, desmoplastic reaction, stellate cells, hyaluronan, CD44

## Abstract

Pancreatic ductal adenocarcinoma (PDAC) is one of the deadliest malignancies. Present-day treatments have not shown real improvements in reducing the high mortality rate and the short survival of the disease. The average survival is less than 5% after 5 years. New innovative treatments are necessary to curtail the situation. The very dense pancreatic cancer stroma is a barrier that impedes the access of chemotherapeutic drugs and at the same time establishes a pro-proliferative symbiosis with the tumor, thus targeting the stroma has been suggested by many authors. No ideal drug or drug combination for this targeting has been found as yet. With this goal in mind, here we have explored a different complementary treatment based on abundant previous publications on repurposed drugs. The cell surface protein CD44 is the main receptor for hyaluronan binding. Many malignant tumors show over-expression/over-activity of both. This is particularly significant in pancreatic cancer. The independent inhibition of hyaluronan-producing cells, hyaluronan synthesis, and/or CD44 expression, has been found to decrease the tumor cell’s proliferation, motility, invasion, and metastatic abilities. Targeting the hyaluronan-CD44 pathway seems to have been bypassed by conventional mainstream oncological practice. There are existing drugs that decrease the activity/expression of hyaluronan and CD44: 4-methylumbelliferone and bromelain respectively. Some drugs inhibit hyaluronan-producing cells such as pirfenidone. The association of these three drugs has never been tested either in the laboratory or in the clinical setting. We present a hypothesis, sustained by hard experimental evidence, suggesting that the simultaneous use of these nontoxic drugs can achieve synergistic or added effects in reducing invasion and metastatic potential, in PDAC. A non-toxic, low-cost scheme for inhibiting this pathway may offer an additional weapon for treating pancreatic cancer.

## 1. Introduction

Pancreatic ductal adenocarcinoma (PDAC), the most aggressive type of pancreatic cancer, is characterized by its very important fibro-inflammatory stromal reaction and low vascularity [1], findings that can be considered pathognomonic for the disease. This feature, known as a desmoplastic reaction, creates a stroma that does not allow the arrival of chemotherapeutic drugs to malignant cells and is one of the fundamental reasons for treatment failures [2]. The desmoplastic reaction entails a fibroblast-rich stroma with an increased dense extracellular matrix (ECM). PDAC stroma provides “protection” to malignant epithelial cells and participates in cancer progression [3]. The intense desmoplastic reaction originates in pancreatic stellate cells (PSCs) that proliferate and produce extracellular matrix proteins, namely fibronectin and collagen creating a particularly dense environment surrounding the malignant cells [4,5]. The unique features of the ECM in pancreatic cancer go beyond “protection” modulating cancer growth through angiogenesis and growth factors [6]. Table 1 shows the evidence that sustains the pro-tumoral concept of the desmoplastic reaction.

Conclusions of Table 1: There are an intense interrelation and cross-talk between PDAC cells and fibroblasts that adopt the form of stellate cells which results in the production of a rich desmoplastic reaction that favors invasion and cancer progression.

Originally, the cancer cell was thought to be the main producer of the ECM [17], however, now it is evident that specialized cells, stellate cells, are the main ECM source. PSCs are in a quiescent stage surrounding pancreatic acini but when malignancy develops, they adopt active participation. A symbiotic relationship between malignant and stellate cells was proposed [9,18,19].

Figure 1 shows a surgically removed PDAC specimen where the relationship between tumor cells (ducts and acini) and the high levels of the intense desmoplastic stroma can be seen.

Recently, Ogawa et al. [20] have distinguished three types of ECMs that differentially influence survival and molecular characteristics. These stromas had different fibroblast populations and transcriptome signatures. Thus, the PDAC matrix is a complex and heterogeneous issue that has direct implications in tumor progression and patient survival [21]. Furthermore, different types of fibroblasts have also been found in the normal pancreas which expands during carcinogenesis with a different prevalence according to subtypes [22].

Interestingly, there are drugs with the ability to inhibit fibroblastic proliferation and their production of collagen and pro-inflammatory cytokines. Pirfenidone is one of such drugs that have been FDA approved for the treatment of pulmonary fibrosis [23]. On the other hand, there are also non-toxic drugs that inhibit the enzymes intervening in hyaluronan production and drugs that decrease the expression of its receptor, CD44.

Targeting pancreatic extracellular environment has been investigated and proposed by many authors [24,25,26,27,28,29,30]. Many “repurposable” drugs with stromal inhibiting abilities have been identified, including metformin [31], all-trans retinoic acid [32], curcumin [33], glutamine analogs such as 6-diazo-5-oxo-l-norleucine (DON) [34], among others.

This manuscript will analyze separately the characteristics of these drugs and finally propose a unified vision of how they should act on the tumor stroma.

We strongly believe that there will be no real breakthrough in PDAC treatment unless the stroma is simultaneously targeted. Pancreatic stroma and tumor should be viewed as one pathological entity in which even if we can separate the parts, functionally they represent a non-divisible unit and so it should be treated.

## 2. Hyaluronan

Hyaluronan (hyaluronic acid), a glycosaminoglycan, is a major component of the extracellular matrix [35,36,37]. Its presence is also prominent in many inflammatory diseases and cancer [38]. Interestingly, when hyaluronan synthesis is pharmacologically decreased in animal models a clear benefit can be achieved in these pathological conditions [39].

Hyaluronan is consistently increased in pancreatic cancer stroma and it exerts a pro-tumoral action [40], including tumor growth [41], proliferation, invasion [42], and metastasis [43,44,45]. Hyaluronan is overexpressed in many inflammatory diseases, [46,47] and certain malignant tumors [48,49].

Hyaluronan consists of a large repeating disaccharide chain (Figure 2).

This polysaccharide has basically two functions:(1)Structural(2)Signaling

As a structural component of the extracellular matrix, it is the main molecule that provides strength and lubrication [48] playing an active role in cell adhesion [50,51] and motility [52,53,54].

As a signaling molecule, it intervenes in proliferation [55,56] and differentiation [57].

Hyaluronan’s structural functions, namely lubrication, and strength, can be fulfilled by this molecule without any receptor. However, signaling and motility require that hyaluronan binds to a specific receptor. The main binding receptor, CD44, is located on the cell surface. There is also another receptor for binding hyaluronan: the intracellular RHAMM (receptor for hyaluronan-mediated motility) While some authors have reported that RHAMM does not seem to have a direct connection with pro-tumoral activities [58], others have found that RHAMM is “an oncogene that regulates signaling through Ras and controls mitogen-activated protein kinase [extracellular signal-regulated protein kinase (ERK)] expression in embryonic murine fibroblasts” [59].

Hyaluronan is produced inside the cell membrane by the enzyme hyaluronan synthase [60]. There are three isoforms of hyaluronan synthase: HAS1, HAS2, and HAS3 (Figure 3 and Figure 4). The production and regulation of each of these enzymes are different. HAS2 produces the longest chain of hyaluronan. HAS2 over-expression produces a hyaluronan that diminishes contact inhibition and promotes cell growth. This was not found with HAS1 and HAS3 over-expression [61]. Therefore, we may assume that HAS2 is the origin of pro-tumoral hyaluronan.

The different molecular weights of hyaluronan are important in cancer [62]: increasing molecular weight (up to 1000 kD) shows increased CD44 binding affinity [63,64].

The chemical structure of hyaluronan is shown in Figure 2 [65], and its interactions in Scheme 1.

## 3. The CD44 Antigen

The CD44 antigen (synonym HCAM) is a glycoprotein acting as an adhesion molecule [68] on the cell surface. Cell adhesion molecules play an important role in cell migration. In fact, CD44 has been shown to be strongly related to invasion [69,70,71] and metastasis [72,73,74]. CD44 has three portions (Figure 5):(1)An extracellular amino-terminal domain to which hyaluronan binds activating signaling. The extracellular domain is also a binding site for other adhesion molecules such as E-selectin and fibronectin.(2)A transmembrane single spanning domain.(3)A short carboxy-terminal intracellular domain responsible for the signaling activity.

Importantly, there are many different isoforms produced by alternative splicing. They all have the ability to bind hyaluronan.

The hyaluronan-CD44 association represents a pro-tumoral hub. Signaling born from the intracellular portion of the HA-activated CD44 activates Ras, MAPK, PI3K [76], and RUNX2-RANKL pathways [77].

Scheme 2 shows CD44 interactions.

This triad, CD44, NHE1, Nav1.5 is present in invadopodia [98,99,100,101], the invasive structure of malignant cells, interacting and promoting invasion and metastasis [102,103].

Nav1.5 promotes invasion through the CD44-src-cortactin signaling pathway [70,104,105,106].

Importantly, CD44 is a marker of cancer stem cells (CSCs) and a regulator of stemness [107,108,109,110], integrating signals between CSCs and pre-metastatic niches [111]. There is evidence that CD44 knockdown can eradicate leukemia stem cells [112]. There is also evidence showing that CD44 positive cells play an important role in gemcitabine resistance in pancreatic cancer [113], distant metastasis, and aggressive behavior [114].

Figure 6 shows the relationship of CD44 with NHE1 and their influence on the cytoskeletal organization, which explains the reasons why CD44 is involved in cellular migration, invasion, and metastasis.

These mechanisms triggering the hyaluronan-CD44 pathway and its protumoral activities can be interrupted by repurposing existing drugs.

In this respect, we have selected two drugs due to their different effects on part of the pathway, and additive or synergistic effects can be expected: bromelain as an inhibitor of CD44 and hymecromone (4-methylumbelliferone) as an inhibitor of hyaluronan. Both drugs are available on the market (in Europe) but are not FDA approved.

### 3.1. Bromelain (BRO)

BRO is an extract from pineapple (*Ananas comosus*) from either the juice of the fruit or from the stem. Actually, it is present in all parts of the freshly picked plant and fruit. The Bromelain extract is a mixture of proteolytic enzymes and other substances including peroxidases.

It has been used for centuries as a folk medicine for the treatment of many different ailments. It is also a meat tenderizer and has found a place in the food industry.

In Europe, it is approved for the removal of necrotic tissues in burn lesions. For the FDA bromelain is only a dietary supplement.

The beneficial effects of bromelain are attributable to its multiple components but mainly to the proteolytic enzymes.

BRO has been used to reduce swelling, especially in ENT surgery and odontological procedures. Although used in folk medicine, the only approved use was issued in 2012 by the European Medicines Agency for removing dead tissue in skin burns. The molecular weight of purified stem bromelain determined by polyacrylamide-gel electrophoresis is 28,500 ± 1000 [122].

In 1972, Gerard was the first author to suggest an anti-cancer activity of BROs [123].

After Gerard, many publications were showed that BROs could play a role in cancer treatment (Table 2).

Furthermore, BRO inhibits/blocks CD44 and reduces its expression on the cell surface but does not reduce its transcriptional output. The relation between BRO and CD44 is shown in Table 3.

BRO was tested on MCF-7 breast cancer cells and the gene profiling showed differences in 1102 genes. The genes with more than 1.5-fold change in expression were: RNA-binding motif, single-stranded interacting protein 1 (RBMS1), ribosomal protein L29 (RPL29), glutathione S-transferase mu 2 (GSTM2), C15orf32, Akt3, B cell translocation gene 1 (BTG1), C6orf62, C7orf60, kinesin-associated protein 3 (KIFAP3), FBXO11, AT-rich interactive domain 4A (ARID4A), COPS2, TBPL1|SLC2A12, TMEM59, SNORD46, glioma tumor suppressor candidate region gene 2 (GLTSCR2), and LRRFIP [148].

At the cellular level, BRO has shown clear anti-tumoral effects and inhibitory activity on CD44. The exact nature of this inhibition is not fully known, but it is not at the transcriptional level. Probably BRO’s proteolytic activity is involved in the mechanism [149].

However, BRO seems to reduce both the CD4+ and CD8+ T- lymphocyte populations and the ratio [150]. Furthermore, BRO also removed CD44 expression from the surface of dendritic cells [151], suggesting that there might be incompatibilities with immunological checkpoint inhibitors. This anti-immune feature may be compensated by BRO’s ability to increase CD2 activation of T-cells [152] in mice. Further research is necessary to find out the exact nature of BRO’s effects on the immune system in humans.

An important question that remains to be answered is: can orally administered BRO achieve therapeutic concentrations at the tumor level?

#### Bioavailability

Oral BRO has been tested in rats [153]. One hour after administration the plasma concentration reached a maximum of 270 mg/mL, which represented 0.003% of the administered dose. In human volunteers, the administration of high doses of BRO resulted in an average plasma concentration of 10 picograms after 48 h [154].

There is clinical evidence that oral BROs have measurable clinical effects.

Heinicke et al. [155] administered oral BRO to 20 volunteers and it decreased platelet aggregation in 17 of them. Another experiment with orally administered BRO in rats showed decreased thrombus formation [156].

The anti-edematous and anti-fibrinolytic activity of BRO was tested in rats that received the medication through an esophageal tube. The antiphlogistic response was proportional to the dose [157].

While the evidence on achieving concentrations necessary for blocking CD44 with orally administered bromelain remains inconclusive, it has been shown that BRO can be absorbed by the human intestine without losing its biological activity [154].

In European markets, BRO formulations between 40 and 750 mg are found. The necessary dose to achieve clinical effects is around 1500 to 2000 mg. These doses are not toxic, and actually, a dose of 12 g/day has been found to lack toxicity [154,158].

### 3.2. 4-Methylumbelliferone (Hymecromone)

Hyaluronan is known to
increase the adhesion and motility of metastatic cells in melanoma [159];increase the motility of pancreatic cancer cell [42] and prostate cancer cells [53];decrease drug delivery to tumors [160,161,162];increase proliferation [163].

Increased hyaluronan expression in tumor stroma is a sign of poor prognosis [164,165,166,167] and there is also evidence of hyaluronan building a pre-niche for driving metastasis [168].

4-methylumbelliferone (4MU) is a hydroxycoumarin that is umbelliferone replaced by a methyl group at position 4. It is an inhibitor of hyaluronan synthase and decreases the production of hyaluronan. Table 4 lists studies showing MU’s anti-tumoral properties.

From Table 4 we can conclude that 4MU has clear anti-tumoral effects by decreasing hyaluronan synthesis.

### 3.3. Pirfenidone

A third drug that could independently target the fibrotic process is pirfenidone.

Pirfenidone is a pyridine that has been approved by the FDA for the treatment of pulmonary fibrosis [191,192,193]. It has also been tested in other fibrogenic diseases such as diabetic nephropathy [194], liver fibrosis [195], and for the normalization of tumor microenvironment increasing chemotherapeutic drugs access to the tumor by reducing hyaluronan and collagen levels in the ECM [196].

Desmoplastic tumors, such as pancreatic cancer, show a very dense ECM that compresses blood vessels [197,198] with two direct effects:increased hypoxia with its pro-tumoral consequences and;inability of chemotherapeutic drugs to reach their target cells.

Importantly, it was found that pirfenidone could block the desmoplastic process in pancreatic cancer [199].

Mechanism of action: Pirfenidone achieves its anti-fibrosis effects through different mechanisms of action. Pirfenidone:(a)inhibits collagen fibrils assembly [200];(b)down-regulates the intercellular adhesion molecule-1 (ICAM1) [201];(c)decreases transformation grow factor-beta (TGFβ) [202] at the translational level [203];(d)down-regulates the pro-fibrotic hedgehog signaling pathway [204];(e)decreases fibroblast proliferation [205];(f)blocks myofibroblast differentiation [206];(g)suppresses tumor necrosis factor alpha (TNFα) [207];(h)decreases cell migration-inducing and hyaluronan-binding protein [208];(i)inhibits MUC1 [209].

In summary, pirfenidone is a small molecule that reduces fibrogenesis and shows anti-inflammatory effects, for which it has been proposed as a therapeutic drug for advanced COVID-19 infection [210].

MUC1 is highly expressed in pancreatic cancer and it was associated with invasion, metastasis, and unfavorable overall survival [211], and inducing epithelial-mesenchymal transition [212]. Many authors consider MUC1 as a valid target in pancreatic cancer treatment [213,214,215,216,217]. Therefore, the ability of pirfenidone to block MUC1 phosphorylative activation is an added benefit of this drug.

Specifically, in pancreatic cancer, pirfenidone has shown the ability to decrease the production of the dense ECM and suppressing cancer cell proliferation [218] and cancer-associated fibroblasts proliferation [219] when associated with cisplatin.

All these characteristics make pirfenidone the best choice to associate with an anti-ECM treatment.

## 4. Discussion

Phytochemicals have been proposed as modulators/inhibitors of the desmoplastic reaction in PDAC in many previous publications (recently reviewed by Ramakrishnan et al. [220]).

Hyaluronan, a glycosaminoglycan, is a normal component of the extracellular stroma. The structural functions of this polydissacharide do not need a receptor and are fulfilled through its physical-chemical properties. PDAC is one of the tumors that usually contains large amounts of hyaluronan [221] and it has been shown that this cancer is highly dependent on hyaluronan production and signaling [182].

Indeed, hyaluronan also shows signaling abilities with a clear pro-tumoral effect and these signaling abilities require the binding to a cell surface protein: CD44. The association of hyaluronan with CD44 represents a pro-tumoral signaling hub. Inhibiting the CD44-hyaluronan association decreases cell motility [222] and probably multidrug resistance (MDR) gene expression [223].

Fortunately, both hyaluronan and CD44 can be down-regulated by existing, non-toxic, and low-cost drugs. BRO, obtained from the pineapple, decreases CD44 on the cell surface through posttranscriptional mechanisms that are not fully known. The most probable mechanism is through BRO’s proteolytic activity on the extracellular domain of CD44. 4MU, on the other hand, acts as an inhibitor of hyaluronan synthase 2, thus reducing the production of hyaluronan.

Both drugs have shown independently anti-tumoral effects. Here, we propose the association of these two drugs to synergistically reduce hyaluronan-CD44 signaling. To the best of our knowledge, this association was never tested either in the laboratory or in the clinical setting.

Figure 7 shows the mechanism of action of the 4MU and BRO association and Table 4 shows many examples of interesting results obtained in pancreatic cancer by inhibiting the hyaluronan-CD44 pathway.

A third drug, pirfenidone, targets the myofibroblasts responsible for the production of the unique PDAC ECM and would complement the effects of the other two.

Here we propose a triple association of three low-toxicity drugs, namely bromelain, 4-methylumbelliferone and pirfenidone to be added to classical pancreatic cancer treatment schemes. The fundamentals behind the triple association are that the three drugs act on different levels of the desmosomal reaction, thus an additive and possibly synergistic effect can be expected from their combined use. Reducing the ECMs pressure will allow less hypoxia and better arrival of cytotoxic drugs to the tumor’s core.

## 5. Conclusions

The hyaluronan-CD44 axis plays two essential roles in cancer: it creates a dense ECM in association with collagen and integrins and forms part of signaling pathways that participate in cellular motion, invasion, and metastasis. Both functions of the axis deserve targeting. Attacking the matrix through the hyaluronan-CD44 axis will not only allow better drug access to the malignant cells but also will reduce invasion and metastasis.

Patients with PDAC, with its dense stroma and high invasive ability, would be particularly benefited by inhibiting collagen production and blocking CD44 activation by hyaluronan.

The hypothesis that synergistic anti-tumoral effects can be achieved by associating bromelain, 4-methylumbelliferone, and pirfenidone to classical pancreatic cancer treatments is presented here. The fundamentals and evidence gathered on these drugs were summarized. BRO and 4MU lack toxicity and pirfenidone have low toxicity, thus we presume that complementing classical treatments with this association would not add to toxicity and can increase treatment effectiveness. We believe that this non-toxic, low-cost scheme for inhibiting this pathway may offer an additional weapon for treating pancreatic cancer.

## Data Availability

Data is contained within the article.

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
