# Peer review of "Targeting the Stromal Pro-Tumoral Hyaluronan-CD44 Pathway in Pancreatic Cancer"

_ijms, 2021, doi:10.3390/ijms22083953_

Round 1

Reviewer 1 Report

Koltai and Reshkin et al. provided new targeting aspects against hyaluronan-CD44 pathway in pancreatic cancer. While their approach is interesting by counting different agents such as 4-methylumbelliferone, Bromelain and Pirfenidone to target this axis, they have also provided numerous literature about this agents. 

However, my first concern is about the way of criticising this phenomenon in pancreatic cancer. Stromal targeting approaches are a long-lasting problem for pancreatic cancer. We are still not sure about the real function of desmoplastic reaction in pancreatic cancer in terms of being tumor promoter or suppressor. Of course hypothesising therapeutic approaches for stromal compartment makes sense, but there should be always more insights before coming to this point. Thus, there are some points to be improved in this hypothetical approach: 

1) Authors should show PDAC structures in Figure 1. They should provide lower magnifications to see whole picture in the PDAC tissue. 

2) So, if they are talking about the components of stromal reaction of pancreatic cancer, they should include other existing literature which has been considered as a therapeutic approach (Shh etc.) as well, addition to CD44.

3) Before talking about the literature of new therapeutic regimens, all of the clinical trials should be mentioned systematically to see the main obstacles in front of stromal targeting concept. I do not see relevant information explaining why first trials were failed to show survival difference considering PEPGH20&Halozyme trials. So the problem should be shaped and analysed and then they can build their hypothesis. Otherwise, in pancreatic cancer and its tumor microenvironment research field can write thousands of hypotheses. 

4) The function of Hyaluronan should be extended in terms of mutational background of pancreatic cancer including Kras and p53.

5) Authors should include some aspects regarding CSCs&CD44 in pancreatic cancer.

6) And please try to group pancreatic cancer studies under one section. Then second section would be other cancer types related research. 

In overall, I will be happy to see these changes and more clinical background for this problem addition to biological background. Therefore in the discussion, these points would be used. 

Reviewer 2 Report

Authors present the hypothesis in which the association of bromelain, 4-methylumbelliferone and pirfenidone with classical treatments can lead to a novel antineoplastic therapy for pancreatic cancer. 

Despite the message is exposed clearly, the overall structure of the manuscript is full of typos and results difficult to follow. My major concern in the discussion section is the misleading "synergism" of compounds. Sometimes it seems that authors refers to additive effect instead of synergic. Please revise deeply the discussion and conclusion taking into account the different interactions that drugs might have. Moreover it would be helpful to move figures and tables from discussion to introduction and dedicate more time on the possible drawbacks of the adjuvant treatment 

Reviewer 3 Report

Koltai et al., proposed a hypothesis entitled “Targeting the pro-tumoral hyaluronan-CD 44 pathway in pancreatic cancer.” The authors explained the role of hyaluronan interactions with CD44 in the context of pancreatic cancer. 

First of all, the content is not well-formatted in the manuscript.

Line 63-67: This reference deserves a citation here. https://doi.org/10.1016/j.apsb.2019.11.008

Please maintain the word's uniformity, e.g., HAS 1 or HAS1; MMP9 or MMP 9; bromelain OR BRO; etc.

Line 122: “chondroitin sulfate that with hyaluronan” --------“chondroitin sulfate that interacts with hyaluronan”

Line 138: Be precise, cytoplasmic domain directly promotes gene expression or activates other molecules, which does this job.

Discussion is not in the order how the other body of the text presented, please rearrange it.

Round 2

Reviewer 2 Report

I don't have any other suggestions